# Changes in DNA Methylation in Response to 6-Benzylaminopurine Affect Allele-Specific Gene Expression in *Populus Tomentosa*

**DOI:** 10.3390/ijms21062117

**Published:** 2020-03-19

**Authors:** Anran Xuan, Yuepeng Song, Chenhao Bu, Panfei Chen, Yousry A. El-Kassaby, Deqiang Zhang

**Affiliations:** 1National Engineering Laboratory for Tree Breeding, College of Biological Sciences and Technology, Beijing Forestry University, No. 35, Qinghua East Road, Beijing 100083, China; xuananran@bjfu.edu.cn (A.X.); yuepengsong@bjfu.edu.cn (Y.S.); BuChenhao@bjfu.edu.cn (C.B.); PanfeiChen@bjfu.edu.cn (P.C.); 2Key Laboratory of Genetics and Breeding in Forest Trees and Ornamental Plants, Ministry of Education, College of Biological Sciences and Technology, Beijing Forestry University, No. 35, Qinghua East Road, Beijing 100083, China; 3Department of Forest and Conservation Sciences, Faculty of Forestry, Forest Sciences Centre, University of British Columbia, Vancouver, BC V6T 1Z4, Canada; y.el-kassaby@ubc.ca

**Keywords:** allele specific expression, 6-BA, DNA methylation, long noncoding RNA, siRNA, poplar

## Abstract

Cytokinins play important roles in the growth and development of plants. Physiological and photosynthetic characteristics are common indicators to measure the growth and development in plants. However, few reports have described the molecular mechanisms of physiological and photosynthetic changes in response to cytokinin, particularly in woody plants. DNA methylation is an essential epigenetic modification that dynamically regulates gene expression in response to the external environment. In this study, we examined genome-wide DNA methylation variation and transcriptional variation in poplar (*Populus tomentosa*) after short-term treatment with the synthetic cytokinin 6-benzylaminopurine (6-BA). We identified 460 significantly differentially methylated regions (DMRs) in response to 6-BA treatment. Transcriptome analysis showed that 339 protein-coding genes, 262 long non-coding RNAs (lncRNAs), and 15,793 24-nt small interfering RNAs (siRNAs) were differentially expressed under 6-BA treatment. Among these, 79% were differentially expressed between alleles in *P*. *tomentosa*, and 102,819 allele-specific expression (ASE) loci in 19,200 genes were detected showing differences in ASE levels after 6-BA treatment. Combined DNA methylation and gene expression analysis demonstrated that DNA methylation plays an important role in regulating allele-specific gene expression. To further investigate the relationship between these 6-BA-responsive genes and phenotypic variation, we performed SNP analysis of 460 6-BA-responsive DMRs via re-sequencing using a natural population of *P. tomentosa,* and we identified 206 SNPs that were significantly associated with growth and wood properties. Association analysis indicated that 53% of loci with allele-specific expression had primarily dominant effects on poplar traits. Our comprehensive analyses of *P. tomentosa* DNA methylation and the regulation of allele-specific gene expression suggest that DNA methylation is an important regulator of imbalanced expression between allelic loci.

## 1. Introduction

Cytokinins are an important class of phytohormones whose discovery was based on their ability to promote cell division in tobacco tissue [1]. These phytohormones, which are synthesized at the root tip [2,3], play various roles in regulating plant growth, development, and differentiation, including meristem function, leaf senescence, source/sink relationships, and vascular development [4,5,6]. The first synthetic cytokinin, 6-benzylaminopurine (6-BA), was produced in 1952. Cytokinins play crucial roles in regulating plant growth and development, including delaying senescence and improving the quality of chlorophyll-containing vegetables by inhibiting chlorophyll degradation [7,8,9]. Physiological and photosynthetic traits are commonly used as indicators to measure the changes of plants in response to external phyhormones [10,11]. The mechanism of these changes in plants needs to be further explored by molecular level variation. However, to date, the responses of woody plants to 6-BA have not been systematically studied at the molecular level, including epigenetic and transcriptional analyses.

DNA methylation is an important epigenetic modification that plays crucial roles in regulating genomic functions. For example, the methylation of microRNA genes regulates gene expression during bisexual flower development in andromonoecious poplar [12,13]. In addition, epigenetic modifications (primarily DNA methylation) affect plant growth and development by controlling flowering time [14]. DNA methylation regulated genomic functions mainly by affecting gene transcription. Zilberman et al. [15] indicated that genic transcription and DNA methylation are closely interwoven processes in *Arabidopsis thaliana*. In addition, previous studies showed that DNA methylation is highly responsive to phytohormones in the ovules female-sterile line [16]. The shoot apex methylome is regulated by ethylene and affects sex determination in cucumber (*Cucumis sativus*) plants [17]. Therefore, the study of 6-BA responsive transcriptional variation and DNA methylation variation can further explore the regulatory effects of cytokinin on poplar. The analysis of whole-genome methylation patterns in response to 6-BA should uncover the precise, specific expression patterns of various transcription regulatory elements, including genes, long non-coding RNAs (lncRNAs), and small interfering RNAs (siRNAs).

Different transcriptional elements play different roles in plants. Among these, lncRNAs are transcripts longer than 200 nucleotides that appear to have no coding potential [18]. LncRNAs play important roles in many biological processes, such as dosage compensation, epigenetic regulation, and cell differentiation [19]. For example, gibberellin (GA)-responsive lncRNAs are associated with wood properties in *Populus tomentosa* [11]. The 24-nucleotide siRNAs represent crucial links in the epigenetic regulatory network in plants. Changes in 24-nt siRNA levels affect DNA methylation at loci regulated by *RNASE THREE-LIKE PROTEIN 2* (*RTL2*) and are inversely correlated with the steady-state levels of mRNA, thus implicating *RTL2* in the regulation of protein-coding gene expression in *A*. *thaliana* [20]. Similarly, changes in 24-nt siRNA levels in *Arabidopsis* hybrids likely play an epigenetic role in hybrid vigor [21]. Therefore, exploring the response patterns of *P. tomentosa* to 6-BA treatment and the interactions between various genetic regulatory elements and their effects on DNA methylation are of vital importance. The relationship between DNA methylation and other epigenetic modifications remains to be fully investigated.

Exogenous phytohormone treatment affects the diploid plant poplar at the transcriptional level not only by altering gene expression but also by influencing the expression of alleles, as allelic differences also play a critical role in gene regulation. Allelic differential gene expression or allele-specific expression (ASE) is commonly observed in the human genome [22] and often reflects the presence of putative allele-specific cis-acting factors of genetic or epigenetic origin. For example, in the human liver, DNA methylation influences the ASE of *CYP1A2* [23]. The variation in ASE levels in heterozygous loci in an individual likely is due to cis-acting polymorphisms, leading to different levels of mRNA expression [24]. The effect of cis-acting elements on gene expression in plants could be explored by identifying ASE under 6-BA treatment.

Here, we identified the changes of physiological and photosynthetic characteristics in *P. tomentosa* leaf in response to 6-BA treatment. To explore the molecular mechanism of these changes in response to short-term 6-BA treatment, we systematically identified 6-BA-responsive variation patterns in *P. tomentosa* leaf at the genome-wide scale, including transcriptional variation (transcriptional expression and allele-specific expression) and epigenetic variation (DNA methylation). Also, 6-BA-responsive differentially methylated regions (DMRs) were further identified. Through annotating DMRs to the genome, we selected a variety of transcription elements located in DMRs that are potentially regulated by DNA methylation, including genes, lncRNAs, and siRNAs. We also performed the first systematic survey of ASE in poplar under 6-BA treatment. The analysis of genomic variation patterns in response to 6-BA treatment facilitate exploring the molecular mechanism of 6-BA responsive traits in poplar and the regulatory mechanism of different genomic elements. On the other hand, the association analysis strategy was used to explore the regulation of genetic variation in DMRs on phenotypic traits of poplar. We then identified SNPs in the DMRs in a natural population of *P. tomentosa*, tested the influence of allelic variation in DMRs on growth and wood properties through association analysis, and dissected their additive and dominant effects. One lncRNA (TCONS_00053467) contained multiple loci with prominent dominant effects. Its target gene (*Potri.002G258000*) is homologous to MALE DISCOVERER1-INTERACTING RECEPTOR LIKE KINASE 2 (MIK2) in *A. thaliana*, which encodes an important regulator of responses to cell wall damage triggered by inhibition of cellulose biosynthesis. *Potri.002G258000* is likely involved in the response of poplar to exogenous 6-BA. The results of this study increase our understanding of the effects of 6-BA-responsive DNA methylation on allele-specific gene expression in *Populus* and lay the foundation for further research on the regulatory effects of 6-BA on plant growth.

## 2. Results

### 2.1. 6-BA Treatment Affects the Physiological and Photosynthetic Characteristics of Poplar

To explore whether 6-BA affects the growth and development of *P*. *tomentosa*, we investigated its short-term effects on physiological characteristics in leaves harvested at 0, 3, 6, 12, and 24 h of treatment. The physiological characteristics included total protein content, peroxidase (POD) activity, sucrose phosphate synthase (SPS) activity, and malondialdehyde (MDA) (Figure 1A–D). At 6 h of treatment, the 6-BA treated group showed significantly higher total protein content and POD activity than the control (increases of 29.94% and 37.62%, respectively). At 3 and 6 h of treatment, the 6-BA treated group showed significantly lower SPS activity than the control (3.04- and 1.39-fold downregulation, respectively). At 12 h of treatment, the 6-BA treated group showed significantly lower MDA than the control. These results indicate that antioxidant enzyme activity significantly increased at 6 h after 6-BA treatment, and cellular damage occurred after 12 h of treatment. The changes of physiological characteristics showed dynamic fluctuations within 24 h, peaked at 6 h, then gradually decreased and tended to flatten. Moreover, they indicate that 6 h of treatment is a key time point to explore the response patterns of poplar to 6-BA treatment; thus, the 6 h time point was selected for subsequent analyses.

Since the changes of physiological indicators were observed to increase first and then decrease, the change pattern of poplar undergoing long-term 6-BA treatment led to our exploration. To further investigate the long-term effects of 6-BA on *P. tomentosa*, we sprayed plants with 6-BA solution once a week for a month and measured the photosynthetic characteristics under short-term (one day) and long-term (one month) treatment. These photosynthetic characteristics included net photosynthetic rate (Pn), stomatal conductance (Gs), transpiration rate (Tr), and intercellular CO_2_ concentration (Ci). Long-term treatment with 6-BA significantly reduced several of these values, with Pn, Gs, and Tr decreasing by 25.98%, 50.37%, and 37.58%, respectively, compared to the control, whereas Ci remained stable (Figure 1E–H). Water use efficiency (WUE = Pn/Tr) increased by 12.22% (Appendix A). We found that the long-term treatment of 6-BA had a significant effect on the photosynthetic characteristics of poplar.

### 2.2. Identification of 6-BA-Responsive Genes and lncRNAs in P. Tomentosa

RNA sequencing was performed on the leaves of the 6-BA group and control group at 6h after 6-BA treatment. Based on the RNA-seq data (Appendix A), we identified differentially expressed genes (DEGs) between treatments based on normalized Fragments Per Kilobase of exon model per Million mapped reads (FPKM) values. A total of 339 genes were differentially expressed between the 6-BA-treated and control groups (|log_2_(fold change)| ≥ 1; and *p* < 0.05; FPKM > 5 at all samples), with 113 and 226 genes up- and downregulated, respectively (Appendix A, Figure 2A). To explore the biological functions of 6-BA-responsive genes, we then characterized these genes functionally using gene ontology (GO) analysis (Figure 2C, *p* < 0.005, FDR < 0.01). We found that 226 down-regulated genes were enriched in 33 GO items (Appendix A). Among these, 25 GO entries are related to biological functions, including ‘nucleoside metabolic process’, ‘aromatic amino acid family biosynthetic process’, ‘dicarboxylic acid metabolic process’, and so on. For categories based on molecular function, the genes were classified into 8 GO categories. The three most overrepresented GO terms were ‘oxidoreductase activity’, ‘catalytic activity’, and ‘transferase activity, transferring acyl groups’. Moreover, up-regulated genes were enriched in 18 GO items (Appendix A), including 14 GO items related with biological process related items and 4 molecular function related items. The main GO terms identified as enriched for the up-regulated category of genes associated with ‘metabolic process’, ‘catalytic activity’, and ‘cellular process’, including ‘cellular macromolecular complex assembly’, ‘cellular carbohydrate metabolic process’, and ‘cellular component assembly’. We analyzed all GO entries of all DEGs and found that the proportion of ‘catalytic’ and ‘metabolic process’ genes was much higher than that of poplar genome. This means that 6-BA treatment had an impact on metabolism and catalytic processes of poplar. The full list of the genes associated with the main enriched GO terms was given in Appendix A.

Genome-wide systematic analysis of lncRNAs revealed 6351 and 4875 lncRNAs (FPKM >1) from libraries from the control and 6-BA-treated groups, respectively. These 6283 lncRNAs, which were stably expressed (FPKM > 1 in at least one group, FPKM > 0 in the other group), included 34 antisense lncRNAs, 69 intergenic lncRNAs, 19 intron lncRNAs, and 6161 sense lncRNAs (Appendix A). An investigation of the basic genomic characteristic of these 6161 lncRNAs indicated that their transcript lengths ranged from 201 to 6706 nucleotides, with a median of 837 nucleotides, which are shorter than the transcripts of protein-coding genes of *P. tomentosa* (median length of 1800 nucleotides). In terms of expression levels, the lncRNAs showed fewer average counts (FPKM = 8.29) than the protein-coding transcripts (FPKM = 18.45). Of the 262 differentially expressed lncRNAs (|log_2_(fold change)| ≥ 1; and *p* < 0.05) selected from the control and 6-BA treated groups, 116 were upregulated and 146 were downregulated (Appendix A, Figure 2B).

### 2.3. Identification and Characterization of 6-BA-Responsive 24-nt siRNAs in P. Tomentosa

We constructed two small RNA libraries from the control and 6-BA-treated groups. After restrictive filtering of contaminating reads, an aggregate of 125,017,531 and 76,823,539 clean reads were obtained, respectively (Appendix A). We identified a substantial number of 24-nt siRNAs tags by performing BLAST analysis of the small RNA sequencing reads to the Pln24NT database (see Appendix A for details). Regions containing a substantial number of 24-nt siRNAs tags were defined as 24-nt siRNA clusters. For each treatment group, the 24-nt siRNA tags within these clusters were standardized to reads per million. Genomic regions associated with 24-nt siRNA clusters will be referred to as 24-nt siRNA loci hereafter. Based on annotation with the Pln24NT database, we identified 7,494,036 and 6,545,777 unique tags of 24-nt siRNAs in the control and 6-BA-treated groups, which were derived from 76,690 and 76,072 clusters, respectively (24-nt siRNA clusters merged into 150 bp windows). The two treatment groups significantly differed in terms of the siRNA size classes of unique and total mapped reads. After 6-BA treatment, both the total number of tags and number of unique tags of 24-nt siRNAs decreased, with the number of unique tags decreasing by 45.29%. These findings indicate that the diversity and abundance of 24-nt siRNAs in poplar sharply decreased under 6-BA treatment.

Through expression analysis, we identified 32,836 and 33,101 reliably expressed 24-nt siRNAs in the control and 6-BA-treated groups, respectively. A total of 69,172 clusters occurred in both groups, with only 15,793 (23%) showing significant differences in siRNA levels (|log_2_(fold change)| ≥ 1; *p* < 0.05), with 38 (0.24%) and 15,755 (99.76%) up- and downregulated, respectively (Appendix A). Among the siRNA clusters with significantly different levels, 1206 were located in 1129 protein-coding genes, including 3′ UTR, 5′ UTR, and coding sequence (CDS) regions, whereas most occurred in intergenic regions (Appendix A). The underrepresentation of TEs (90.61%) predominantly occurred in genic regions, whereas the overrepresentation of TEs (50.08%) occurred in intergenic regions (Appendix A). Moreover, 99.76% of the significant 24-nt siRNA clusters were downregulated after 6-BA treatment.

### 2.4. Identification of 6-BA-Responsive Allele-Specific Expression Loci in P. Tomentosa

As a diploid plant, poplar contains two sets of genomes, which originated from parents, respectively. Therefore, there will be a small number of allele variation loci (heterozygous loci), resulting from differences between father and mother on a pair of homologous chromosomes. To explore the differential expression of alleles in poplar in response to 6-BA treatment, we used RNA-sequencing reads mapped to the reference genome to identify loci displaying differential expression between two alleles in the whole genome and genotyped samples from the 6-BA-treated and control groups (Figure 3C). Allele-specific expression (ASE) analysis was conducted for ASE loci identified that were expressed at detectable levels in the RNA-sequencing reads. Among these loci, 105,913 (15.61%) were distributed over 19,200 genes, with 1 to 22 heterozygous loci per gene (average 3.8); among these, 78.19% were located in annotated exons (Figure 3C). We detected 102,819 loci in 19,200 genes showing differences in ASE levels after 6-BA treatment (Appendix A). To explore the variation in ASE of the 6-BA-responsive genes, we divided these genes into two groups: up- and downregulated genes. We used A/R to calculate the ASE level: A refers to the number of reads that contained the alterative allele, and R refers to the number of reads that contain the reference allele. As shown in Figure 3B, under 6-BA treatment, the A/R value (number of alternative loci/number of reference loci) of the upregulated genes decreased, whereas that of downregulated genes increased, indicating that 6-BA treatment had different effects on genes with different expression patterns. These results indicate that 6-BA treatment causes a universal imbalance in the expression of alleles, as these values significantly deviated from a 1:1 ratio. Furthermore, most differential expression was detected in the exons of genes, thereby playing important roles in regulating gene expression and the translation of proteins controlling growth in *P. tomentosa*.

### 2.5. Variation of DNA Methylation in P. Tomentosa Under 6-BA Treatment

Based on the observed changes in physiological and photosynthetic traits, we used leaves from control and 6-BA-treated plants at 6 h of treatment for whole-genome bisulfite sequencing (WGBS) (Appendix A). For the control group, ~82 million clean reads were generated, and after filtering low-quality and duplicate reads, 28,762,783 reads were mapped to the *P. trichocarpa* genome (version 3.0) (~34.81%). A total of 3.22% of the cytosine sites were methylated in the whole genome. These sites were classified as being in the CG, GHG (with H representing A, C, or T), or CHH contexts, representing 32.91%, 30.63%, and 36.45% of the total methylated cytosine sites, respectively. Among 6-BA-treated plants, the mapping percentage was approximately 34.12%. After removing low-quality and duplicate reads, ~74 million uniquely mapped high-quality reads were obtained, with 25,367,876 reads mapped to the *P. trichocarpa* genome (version 3.0). A total of 3.22% of the cytosine sites were methylated in the whole genome, including of 31.14% (CG), 29.05% (CHG), and 39.81% (CHH) (Figure 4A). Therefore, the total proportion of methylated cytosines sites did not change under 6-BA treatment, whereas the proportion of methylated cytosines in different contexts changed significantly. These results indicate that the response of *P. tomentosa* to the external environment at the epigenetic level is not only reflected in the number of methylated cytosines sites, but it is also related to the different contexts of methylation, which originate from different biological processes. The proportion of CHH methylation decreased after 6-BA treatment, while the proportion of methylation in the other contexts (CG and CHG) increased. CHH methylation is thought to be related to RNA-mediated DNA methylation (RdDM), which is mediated by 24-nt siRNAs, suggesting that 6-BA-responsive siRNAs might be involved in this progress. Moreover, the variation in methylation levels in different genomic features was consistent in the control and 6-BA-treated groups: both showed peak methylation levels in the promoter region close to the 5′ UTR of the gene, and the methylation level gradually decreased with increasing proximity to the exon region (Appendix A). The distributions of methylation levels in different contexts in the control (CK) and 6-BA-treated groups are shown in Figure 4C. To examine the responses of DNA methylation, we also measured the expression of genes encoding methylation-related enzymes. DNA methylation in plants mainly depended on four cytosine methyltransferases, including *methyltransferase 1* (*MET1*), *domains rearranged methyltransferase* (*DRM*), *chromomethylase* (*CMT*), and homologue of *DNA methyltransferase 2* (*DNMT2*) [25,26,27,28]. Nine DNA methyltransferases-coding genes were identified in poplar from the KEGG (http://www.kegg.jp/kegg/pathway.html) database. Here, we profiled the expression of 9 DNA methylation-related genes under 6-BA treatment (Appendix A). Among these, 77.78% (7 of 9) of genes were significantly influenced by 6-BA treatment in the short-term. The expression levels of two *DRM2* genes (*Potri.001G347000* and *Potri.014G049500*) was significantly up-regulated under 6-BA treatment, which maybe resulted in the increase in proportion of CHH methylated cytosines. These results indicated that DNA methylation is influenced by exogenous 6-BA treatment.

DNA methylation is a dynamic process in the plant genome. It has been proven that the DNA methylation status of parents can be passed on to their offspring. Therefore, the DNA methylation status of alleles derived from parental inheritance may be different. We then genotyped the methylated CpG sites at the allele level. Our experimental materials come from a natural germplasm bank, and poplars are highly heterozygous, which meant it was impossible to acquire the genome of parents. So, we defined the parameter S, which represented the methylation support of loci (S = number of mC reads/(number of mC reads + number of C reads in single locus) to reflect the difference in allelic methylation levels. In total, 4,285,665 CpG loci were found simultaneously in the 6-BA group and the control group. By calculating the methylation support of these loci, it was found that 39.7% of the CpG sites had an obvious change of allelic methylation levels (|(S_6-BA_-S_CK_)|/S_CK_ > 0.5)) under 6-BA treatment. These results indicated that nearly half of CpG sites exhibited 6-BA-responsive changes in allelic methylation levels (Appendix A).

### 2.6. Variation of Differential Methylation Regions (DMRs) Under 6-BA Treatment and ASE Analysis of Transcriptional Elements Within in DMR Boundaries

To systematically explore the epigenetic response of poplar to 6-BA treatment, we identified and annotated DMRs based on bisulfite sequencing data. A total of 460 DMRs with differences in magnitude between the control and 6-BA treated groups were defined as DMRs (*p* < 0.05) (100–1895 bp long (Appendix A)). We mapped the DMRs to the *P. trichocarpa* genome (version 3.0) based on location information and found that they overlapped with various genomic elements, including protein-coding genes, lncRNAs, and 24-nt siRNA cluster sequences (24-nt siRNA database from Pln24NT). Specifically, we identified 180 protein-coding gene sequences, 17 lncRNA coding sequences, and 59 24-nt siRNA cluster sequences within the boundaries of these DMRs (Figure 4B). Most DMRs contained only a single element sequence, with 163 DMRs containing single protein-coding gene sequences, 14 containing single lncRNA sequences, and 41 containing single 24-nt siRNA sequences. The remaining DMRs contained two genome elements, including 18 DMRs simultaneously occurring in protein-coding genes and siRNA sequences and one DMR simultaneously occurring in lncRNA and siRNA (Figure 4B, Appendix A). Transposable elements were present in 79.34% of the 24-nt siRNA clusters within DMR boundaries (Appendix A).

To further explore the effects of DNA methylation on transcriptional regulation and ASE in poplar, we performed expression analysis and ASE analysis of transcription elements located in DMRs, which could potentially be regulated by DNA methylation, including 180 protein-coding genes, 17 lncRNA genes, and 59 siRNA cluster genes. Of the six differentially expressed protein-coding genes in response to 6-BA treatment, two were upregulated and four were downregulated (|log_2_(fold change)| ≥ 1, *p* < 0.05). Among the lncRNAs, five were expressed stably under 6-BA vs. control treatment, with fold change values between 0.81 and 3.22. In addition, 37 24-nt siRNAs clusters were differentially expressed under 6-BA treatment, all of which were significantly downregulated (|log_2_(fold change)| ≥ 1, *p* < 0.05, Appendix A).

To further explore the ASE patterns of these genes and lncRNAs in DMRs, we selected two protein-coding genes and two lncRNA genes (one with maximum and the other with minimum fold change values) within DMRs to calculate the ASE levels of SNPs located in the genes. As shown in Figure 5, the ASE levels of both protein-coding genes (*Potri.011G140400* and *Potri.006G239700*) decreased in response to 6-BA treatment, which was accompanied by the downregulated expression of *Potri.011G140400* and the upregulated expression of *Potri.006G239700* (Figure 5A,B). *Potri.011G140400* and *Potri.006G239700* were located in DMRs, where methylation level increased under 6-BA treatment. The ASE levels of lncRNA *TCONS_00082911* and lncRNA *TCONS_00240091* simultaneously increased, with the former showing downregulated expression and the latter showing upregulated expression (Figure 5C,D). The two lncRNAs were located in DMRs, where DNA methylation levels decreased simultaneously. These results indicate that the expression level of protein-coding genes and lncRNA were not directly related to their ASE level, while the variation of DNA methylation level and ASE level were consistent in protein-coding genes and lncRNAs.

### 2.7. Variation in 6-BA-Responsive DMRs is Associated with Phenotypic Variation

We conducted a series of analyses to investigate the functional roles of the 6-BA-responsive elements. GO enrichment analysis showed that the 6-BA-responsive DNA methylated genes were enriched for the categories ‘regulation of cellular metabolic process’, ‘cellular biosynthetic process’, and ‘cellular macromolecule metabolic process’. This enrichment suggests that 6-BA responsive genes are likely involved in phenotypic variation. To test this hypothesis, we conducted association analysis to investigate the potential functional roles of 6-BA-responsive DMRs in phenotypic variation. We selected 412 6-BA-responsive DMRs for SNP analysis by re-sequencing a natural population of *P. tomentosa*. After discarding SNPs with minor allele frequencies <5% and SNPs with >25% missing data, we identified 3604 common SNPs. We used the SNPs within the DMR boundaries to perform association analyses relative to 10 growth and wood property traits, including height (H), diameter at breast height (DBH), stem volume (V), holocellulose (HC), hemicellulose (HEMC), α-cellulose (AC), lignin contents (LC), fiber length (FL), width (FW), and microfibril angle (MFA) (Appendix A and S15). Using MLM in TASSEL5.0, we detected 182 significant associations between 152 SNPs from 412 DMRs and seven growth and wood property traits (DBH, V, HEC, HC, AC, LC, FW) (*p* < 0.001, *Q* < 0.1), with 12.39–27.12% of the phenotype variance (R^2^) explained by each SNP (Table 1, Appendix A).

Since dominant effects results from the interaction between alleles within a gene locus, the value of the dominant effect fully reflects the interaction between alleles. We subjected 102 ASE loci in the 6-BA-responsive DMRs to further analysis. Association analysis showed that 52.94% of the ASE loci prominently exhibited dominant effects on poplar traits (d > a). We identified 24 loci associated with multiple traits (DBH, V, HEC, HC), where dominant effects had a significant advantage over additive effects (d/a > 2, Appendix A). Interestingly, among these loci, four were jointly annotated to the lncRNA *TCONS_00053467* and were correlated with different traits (DBH, V); the ASE level changed significantly, accompanied by dominant effects (*p* < 0.05, Figure 6C,D). Computational prediction identified six potential cis-regulated target genes and six potential trans-regulated target genes for *TCONS_00053467*. Among these target genes, the cis-target gene for *Potri.002G258000* and *TCONS_00053467* partially overlapped in the genome: *Potri.002G258000* was significantly downregulated and *TCONS_00053467* was significantly upregulated under 6-BA treatment (Figure 6B). These genes are located on sense and antisense strands, respectively. *Potri.002G258000* is homologous to *MIK2/LRR-KISS*, a *leucine-rich repeat receptor kinase* (*LRR-RK*) gene in *Arabidopsis* [29,30]. As shown in Figure 6A, multiple interacting elements exist in this candidate region, in which hypermethylated *DMR_Chr02_24663250* partially overlaps with *siRNA_cluster_32657* in the upstream region. The expression of *siRNA_cluster_32657* was significantly downregulated under 6-BA treatment (Figure 6B).

## 3. Discussion

### 3.1. The Responses of Physiological Characteristics and Photosynthetic Indices to 6-BA Treatment in P. Tomentosa

Previous studies have demonstrated cytokinins play important roles in plant growth and development. Leaf spraying is a common method to explore the effects of hormones on plant growth and development. Physiological and photosynthetic indicators can be used as indicators to measure the responsive changes of plants under different treatments. Therefore, the photosynthetic and physiological response patterns of exogenous cytokinins (6-BA) treatment have been studied in many plants. For example, the treatment of 6-BA significantly increased the contents of chlorophyll, soluble protein, soluble sugar, and the activities of SOD and POD in cotton [31]. Zhang et al. [25] found that 6-BA application increased peroxidase (POD) activities and decreased the MDA contents in cottons, which are consistent with the changes in poplar. It was found that 6-BA alleviated chilling injury in cucumber fruit through improving antioxidant enzyme activities and total antioxidant capacity as well as maintaining higher levels of ATP content and energy charge [26]. In this study, we have shown that total protein content and POD activities increased and then decreased, while SPS and MDA activities decreased at the early stage and then increased under 6-BA treatment in poplar. The changes of physiological characteristics peaked at 6 h after 6-BA treatment, suggesting that 6-BA rapidly triggers physiological changes in poplar. In addition, the study on the change patterns of poplar physiological traits under GA treatment has been reported [15]. It was found that the changes in POD and MDA under GA treatment were consistent with those observed under 6-BA treatment. However, the changes in total protein content and SPS activity were opposite those under 6-BA treatment. Interestingly, the changes of physiological characteristics also peaked at 6 h after treatment. Therefore, we reasoned that the physiological responses of *P. tomentosa* to different growth regulators are completely different. However, the responses were similar after 6 h, implying that responses to phytohormones occur rapidly in poplar, although the specific regulatory mechanisms remain unclear. These changes under 6-BA treatment may be caused by exogenous 6-BA affecting the metabolism of related enzymes in the short-term, so the physiological changes increased, while exogenous 6-BA, as stress factors, may trigger the endogenous homeostasis in plants to regulate the physiological indexes returning to normal level. Among these physiological characteristics, POD, a protective enzyme, removes hydrogen peroxide from cells to reduce oxidative damage, thus delaying leaf senescence [32]. POD activity in leaves increased significantly under 6-BA treatment, confirming the effect of cytokinin on delaying leaf senescence. Lipid peroxidation is an inherent feature of leaf senescence [33]. MDA activity in poplar leaves decreased significantly under 6-BA treatment, indicating that lipid membrane peroxidation was enhanced, leading to delayed leaf senescence. Therefore, 6-BA may delay leaf senescence by increasing the activity of photosynthetic and protective enzymes, thereby reducing secondary metabolic activity.

Based on the changes of physiological indicators under short-term treatment, which showed a trend of increasing to the peak and then gradually weakening, we further measured the photosynthetic indicators to explore the effects of long-term treatment on poplar. In this study, Pn, Gs, and Tr significantly decreased in *P. tomentosa* under long-term 6-BA treatment, with Ci increasing slightly. WUE increased under long-term 6-BA treatment. However, this pattern is opposite that reported for cotton, where Pn and Tr significantly increased in response to 6-BA, while WUE remained stable [23]. Ding et al. [24] indicated that under normal light conditions, the foliar application of 6-BA had no effect on any photosynthetic parameter in cucumber plants. Thus, it appears that different species exhibit different photosynthetic patterns in response to 6-BA treatment. We found Pn decreased, while Ci increased slightly under 6-BA treatment, which may be caused by non-stomatal restriction factors related with the limitation of photochemical activity, which hindered the utilization of CO_2_, resulting in the accumulation of Ci and the decrease of Pn. In addition, these changes may be caused by the decrease of Gs, resulting from stress effects of 6-BA treatment. On the other hand, 6-BA short-term treatment caused the slight increase of Tr, while Pn, Gs, and Tr were almost unchanged, which demonstrated that photosynthesis is insensitive to short-term 6-BA treatment and lags behind physiological characteristics. These findings showed that the long-term treatment of 6-BA had definite effects on poplar, so it is of great significance to study the regulatory mechanism of 6-BA on poplar. In this study, it was found that 6-BA can trigger rapid physiological changes in poplar within several hours, and 6 h after 6-BA treatment was a key time point. Therefore, 6 h was selected for further omics analysis to explore the immediate effects of 6-BA treatment on poplar at molecular level in order to analyze the molecular regulatory mechanism of 6-BA. We explored the effects of cytokinin on poplar at the transcriptional, epigenetic, and allelic levels from a molecular perspective through sequencing technology. Exogenous cytokinin produced genome-wide changes, including changes in photosynthetic indicators, enzyme activity, and epigenetic modifications, which were not observed in previous studies, forming the basis for changes in poplar growth.

### 3.2. 6-BA Responsive DNA Methylation in Poplar

Epigenetics refers to genetically stable changes of chromatin states that alter gene expression and phenotypic traits independently of alteration of the DNA sequence, which consist of a wide range of biochemical processes, such as DNA methylation, histone modifications, and small or long non-coding RNAs. Among these, DNA methylation plays an important role in plant genome defense, regulation of gene expression, and influence on plant growth and development [34,35]. In eukaryotes, methylation is mainly involved in gene expression. Previous studies have shown that DNA methylation is a dynamic process in response to environmental changes, including abiotic stress and chemical treatment [30,31,32]. Dowen et al. [36] found that *At1g13470* was significantly demethylated and its expression was up-regulated in *A. thaliana* under salicylic acid treatment. *At1g13470* functions as a like-transposon silencing gene that can inhibit gene expression through RdDM. In addition, the relationship between DNA methylation and gene expression has been extensively studied at the genome-wide level. Liang et al. [37] divided poplar genes into four groups according to their expression levels: high expression, medium expression, low expression, and silencing genes. It was found that the silencing gene had an obvious hypermethylation level compared with the expression genes, indicating that gene silencing might be caused by hypermethylation. The methylation level of silencing genes increased significantly in poplar under drought stress. These studies suggest that DNA methylation can affect gene expression in response to changes in the environment.

Since genic transcription and DNA methylation are closely interwoven processes, we conducted BS sequencing to mine systematically the DNA methylation response pattern under the short-term 6-BA treatment. We screened 339 DMRs by strict thresholds, which were located in the promoter regions of the protein-coding gene, suggesting the potential role of DNA methylation in downstream gene regulation. The expression levels of DNA methylation-related genes changed significantly in response to 6-BA treatment. Previous studies found *Potri.001G002300* was hypermethylated at 6h after indole-3-acetic acid (IAA) treatment in poplar, resulting in alternative splicing that produced a transcript encoding a dysfunctional protein lacking its catalytic center and ATP binding site [38]. These results indicated that DNA methylation is sensitive to external hormone treatment, which is an important mechanism affecting gene expression.

### 3.3. 6-BA-Responsive ASE Analysis in Poplar

Differential gene expression between the two alleles or allele-specific expression (ASE) in the genome is common [39,40], which reflects the presence of putative allele-specific cis-acting factors of either genetic or epigenetic origin. Here, we performed the first systematic survey of ASE in poplar under 6-BA treatment. Since the identification of ASE loci is performed on the whole-genome scale, we faced several difficulties, such as the wide range of locus distribution, large amount of data, and complex loci variation. It is particularly important to choose good identification criteria for ASE loci. Previous studies have used the NS-12 BeadChips assay to measure the average fluorescent signals of single alleles (A1/(A1+A2) or A2/(A1+A2)) in DNA and RNA, respectively. To obtain a quantitative measure of ASE, the authors subtracted the allele fractions (A1/(A1+A2)) measured in DNA from that in RNA and referred to this difference as the ASE level [41]. The advantage of this approach is that it eliminates the effect of genetic variation on ASE; however, this approach is not suitable for ASE analysis of sequencing data.

In the present study, we used bioinformatics to compare RNA-seq data to identify ASE loci and selected stable heterozygous loci for further analysis. To minimize bias in SNP calling, we used a series of measures to increase the accuracy of genome mapping. First, we used software for RNA-specific comparisons in genome mapping, which uses an improved BWT algorithm that allows introns to be identified for cross-intron mapping to the genome. We then used the Picard tool to identify duplicates generated by PCR amplification. We also used the SplitNCigarReads tool in GATK to split the reads into exon segments by removing Ns and hard-clipping any sequences overhanging into intronic regions. Importantly, this analysis relied on the *P. trichocarpa* genome (version 3.0), which to some extent led to mapping bias of RNA-seq data in *P. tomentosa*. Therefore, we selected loci with high numbers of reads (the sum of number of reads containing the alterative allele and number of reads containing the reference allele was greater than 5) to reduce sequencing and alignment errors caused by the use of the *P. trichocarpa* genome (version 3.0). Actually, the reference genome served as an anchor template for indirect comparisons between the 6-BA treatment and control groups. The results should be accurate as long as the locations of loci between the 6-BA treatment group and control group are consistent.

We then performed ASE analysis based on the large-scale allelic variation loci identified by the SNP calling pipeline of RNA-seq. We selected loci from the 3′ UTR with strong specificity in a single gene for genotyping to score the ASE level of that gene. The advantage of this approach is that the non-conservative regions of the 3′ UTR are selected as candidate regions, which are sensitive to external environmental changes and growth regulators. However, due to the complexity of gene transcriptional regulatory elements, it is not clear whether this method can truly represent ASE patterns of genes; this notion requires further exploration. As we demonstrated based on 19,200 genes with identifiable functional variations, allele-specific gene expression was widespread under 6-BA treatment. Such strong, consistent, imbalanced expression likely occurs because one of the alleles is favorable while the other is unfavorable under 6-BA treatment. Therefore, allele-specific expression analysis provides a new perspective for exploring the variation patterns of the transcriptome in response to the external environment. The presence of allele-specific regulation elements, which were influenced by 6-BA in varying degrees, resulted in allelic specific expression. It provides a new way to explore the transcriptional regulation effects of 6-BA.

### 3.4. Effects of DNA Methylation on the Transcriptional Regulation at Allelic Level

The functional consequences of promoter methylation on transcriptional regulation are well known [42], and DNA methylation in the promoter region inhibits gene expression. However, the influence of DNA methylation on gene expression of specific alleles remains largely unresolved, especially in plants. A previous study found that parental allele-specific expression (gene imprinting) is closely associated with DNA demethylation and the siRNA-dependent RdDM pathway, which regulates seed development in maize [43]. Kinoshita et al. [44] found that maintenance of endosperm-specific and parent-of-origin-specific FWA expression depends on maintenance DNA methyltransferase *MET1* in *Arabidopsis*, which indicated that the maintenance of FWA imprinting depends on the maintenance DNA methylation machinery. In this study, we systematically identified the ASE loci to further explore the ASE patterns of transcriptional elements in DMRs. We selected four representative transcriptional elements to analyze the relationship between DNA methylation and ASE. It was found that the variation of DNA methylation level and ASE level were consistent in protein-coding genes and lncRNAs.

On the other hand, allele-specific DNA methylation also influences gene expression. For example, a correlation between ASE and CpG site methylation was detected in bone marrow and blood samples from children with acute lymphoblastic leukemia. The shape of the regression curve for ASE level is a function of methylation level, with a maximum at a beta-value of 0.59, and is consistent with increased methylation of one allele prior to complete methylation [41]. In addition, at the single allele level, Blewitt et al. [45] found that DNA methylation in mice can result in a range of coat colors, from yellow (due to the influence of a single allele) to agouti viable yellow (Avy) (resulting from a retrotransposon insertion upstream of the Agouti gene via hypomethylation of the long terminal repeat). In the current study, we used the methylation support rate to genotype methylation loci and analyzed the changes in methylation at the allele level. We defined the parameter S, which represented the methylation support of loci to reflect indirectly the degree of imbalance in allelic methylation levels. It was found that nearly half of CpG sites exhibited 6-BA-responsive changes in allelic methylation levels, which lays the foundation for further study on the effect of allelic methylation level on gene expression in poplar.

### 3.5. The Relationship Between Dominant-Effect Expression and ASE Analyzed by Association Analysis

We conducted an association analysis of the DMRs to analyze the genetic effects of the regions potentially regulated by DNA methylation. Among these, the dominant effect might be caused by allele-specific or imbalanced expression (the SNPs for two allelic sequences deviated significantly from a 1:1 ratio). The detection of ASE of genes under 6-BA treatment strengthens this hypothesis. Perhaps favorable alleles are expressed at higher levels in response to the growth regulator 6-BA, resulting in dominance. This finding suggests that two alleles can adapt to growth regulators in different ways.

We identified a key region of a DNA sequence (*DMR_Chr02_24663250*) with complex transcriptional regulation involving multiple transcriptional elements, where differential hypermethylation occurred under 6-BA treatment. *Potri.002G258000*, a homolog of *MIK2*, was identified by genetic analysis. This gene was targeted by *TCONS_00053467*, which contains four dominant SNPs associated with tree diameter at breast height (DBH) and stem volume (V). MIK2 is an important regulator of responses to cell wall damage triggered by the inhibition of cellulose biosynthesis [46], supporting the notion that 6-BA treatment promotes cell division through lncRNA-mediating transcriptional regulation.

In addition, we detected a siRNA cluster near this DMR. We further explored the expression levels of the above-mentioned transcriptional elements. The expressions of MIK2 and the nearby siRNA cluster were negatively correlated with *TCONS_00053467* expression. SiRNA is closely associated with de novo methylation [47]. Our results indicate that *siRNA_cluster_32657,* which is located in *DMR_Chr02_24663250*, maintains the hemi-methylated state of this DNA region. Under normal growth conditions, hemi-methylated *DMR_Chr02_24663250* might regulate the ASE of adjacent *TCONS_00053467* expression. Transcription of the lncRNA *SRG1* driven by the promoter of the adjacent *SER3* gene repressed *SER3* expression by interfering with the binding of *Pol II* to its *cis*-element [33]. In the current study, the expression levels of *TCONS_00053467* and its *cis*-target gene were negatively correlated. *TCONS_00053467* partially overlaps (153 bp) with the 3′ terminus of *MIK2* and is located on the antisense strand. The concurrent expression of a pair of genes on complementary strands might induce the biosynthesis of double-stranded RNA (dsRNA), which would trigger RNA interference, resulting in adjacent gene silencing. However, no endogenous siRNAs were detected in this region. Thus, the upregulated *TCONS_00053467* might also repress *MIK2* expression via their shared flanking regions. However, this regulatory mechanism requires further study.

Collectively, the above results suggest that the hemi-methylation of *DMR_Chr02_24663250* might play important roles in the dominant effect of *TCONS_00053467* on D and V through mediating ASE of the *TCONS_00053467-MIK2* module. Under 6-BA treatment, *siRNA_cluster_32657* expression was significantly downregulated, along with hypermethylation of *DMR_Chr02_24663250*, pointing to possible negative feedback regulation between methylation level and the expression of adjacent siRNA clusters. Our study provides a reference for investigating allelic variation and epigenetic variation of various elements in the genome under 6-BA treatment at the molecular level, and it provides a new perspective on the effects of 6-BA on woody plants for future functional studies.

## 4. Materials and Methods

### 4.1. Plant Materials and 6-BA Treatment

*P. tomentosa* (clone number ‘1316′) was used in this study. Explants were collected from ‘1316′, which were used to establish clonal systems through tissue culture. Tissue culture seedlings of ‘1316′ taking root were transplanted into the containers in a mixture of soil, organic matter, vermiculite, and perlite (1:1:1:1), which were grown under a 15h light/8h dark photoperiod in a greenhouse at Beijing Forestry University, Beijing, China (40°0’N, 116°20’E). Thirty one-year-old *P. tomentosa* ramets with the same growth status were selected as experimental materials, which were randomized in two separate groups: treatment (6-BA treated) and control (distilled water). The 6-BA solution preparation method was as follows: 6-BA solid powder (Sigma-Aldrich, St. Louis, MO, USA) was firstly dissolved in a small amount of 1 M of HCl and then diluted in distilled water to the appropriate concentration. All the leaves in the treatment group and control group were sprayed with 6-BA solution and distilled water, respectively. Spraying 125mL of liquid each time until soaking (with drops of liquid dripping down) ensured that each leaf was fully soaked.

First of all, we conducted a preliminary experiment to determine the appropriate concentration of 6-BA by spraying different concentrations of 6-BA on *P. tomentosa* leaves. We selected another 15 ramets (5 concentration × 3 biological replicates) for 6-BA solution spraying with different concentrations, and the concentration gradient of 6-BA solution was 0.1, 1, 10, 100, and 1000 μmol/L. Three biological replicates were set for each concentration treatment. The experiment used the test of least significant difference (LSD) multiple comparisons to compare different 6-BA concentration influences on physiological characteristics. The functional leaves expanded fully (the third to fifth leaves away from the top of brunch) and were harvested from the same position of each ramet for physiological characteristic measurement. The changes of total protein, SPS activity, and MDA content appeared inflection point under 100 μmol/L 6-BA treatment (Appendix A-D). Thus, 100μmol/L was selected as the appropriate concentration of 6-BA treatment.

Then, we sprayed all the leaves of ramets in the treatment group and control group with 100 μmol/L 6-BA solution and distilled water according to the above methods, respectively. For both groups, three functional leaves expanded fully (the third to fifth leaves away from the top of brunch) were harvested from the same position of each ramet at 0, 3, 6, 12, and 24 h after treatment (2 treatments × 5 time points × 3 biological replicates), immediately frozen in liquid nitrogen, and stored at −80 °C until further use. Three different ramets were set up at different time points in both the treatment and control groups as biological replicates to eliminate the effect of deviation in leaf position.

### 4.2. Measurement of Physiological, Growth, and Wood Properties as well as Photosynthetic Indices

Various parameters were measured in plants from both treatment groups (6-BA treated and control) at five treatment times as follows: (i) four physiological traits. Total protein content was measured by the Bradford method using BSA as the standard. Sucrose phosphate synthase (SPS) activity was detected using a Sucrose Phosphoric Acid Synthase Assay Kit (Nanjing Jiancheng Bioengineering Institute, Jiangsu Province, China). Peroxidase (POD) activity was measured using a Plant POD Assay Kit (Nanjing Jiancheng Bioengineering Institute). Malondialdehyde (MDA) content was analyzed according to the method of (Heath and Packer., 1968); (ii) three growth traits, including height (H), diameter at breast height (DBH), and stem volume (V), which followed standard procedures described in Zhang et al. [48]; (iii) seven wood property traits, including holocellulose (HC), hemicellulose (HEMC), α-cellulose (AC), lignin contents (LC), fiber length (FL), width (FW), and microfibril angle (MFA). LC was derived by wet chemistry analyses as described in Porth et al. (2013). The detailed methods for sampling and measuring the three growth traits (DBH, H, and V) and seven wood property traits are described in Du et al. [49]; and (iv) four photosynthetic indices, including net photosynthetic rate (Pn), stomatal conductance (Gs), transpiration rate (Tr), and intercellular CO2 concentration (Ci); these parameters were measured from fully expanded leaves (three functional leaves, i.e., the top four to six leaves) using the LI-6400 Portable Photosynthesis System (LI-COR Inc., Lincoln, NE, USA) following the manufacturer’s instructions at 9:00–11:00 AM on sunny days. Each leaf measurement was performed using three replications. WUE was calculated by the formula WUE = Pn/Tr.

### 4.3. DNA Extraction and Bisulfite Sequencing

At 6 h after treatment, leaves of the 6-BA-treated and control groups were collected for bisulfite sequencing using three biological replicates for each treatment time. Total genomic DNA was extracted from young leaves using a DNase Plant Mini Kit (Qiagen China, Shanghai, China) following the manufacturer’s protocol. The DNA was sonicated to generate fragments with a mean size of approximately 250 bp, followed by DNA repair of blunt ends (3′ ends) by the addition of dA and adaptor ligation. Bisulfite modification and conversion of genomic DNA were then performed using an EpiTect Bisulfite Kit (Qiagen, Valencia, CA, USA) according to the manufacturer’s instructions. The resulting DNA from 6-BA-treated and control plants for all treatment time points was subjected to paired-end sequencing with a read length of 101 nt for each end using the ultrahigh-throughput Illumina Hiseq2000 platform as per the manufacturer’s instructions.

### 4.4. RNA Extraction and RNA-Sequencing

At 6 h of treatment, leaves of 6-BA-treated and control plants were subjected to RNA sequencing using three biological replicates per treatment time. Total RNA was extracted from the samples using a Qiagen RNase Kit (Qiagen China, Shanghai, China) according to the manufacturer’s instructions, followed by on-column DNase digestion to purify the RNA samples using an RNase-Free DNase Set (Qiagen). The RNAs were assessed with a NanoDrop ND-1000 and Agilent Bioanalyzer 2100 before being used to construct strand-specific RNA-seq libraries as described in the TruSeq RNA sample preparation guide. The strand-specific libraries were sequenced using an Illumina HiSeq 2500 instrument in which 100-nucleotide paired-end reads were produced. Library construction and sequencing were performed by Shanghai Biotechnology Corporation (Shanghai, China).

### 4.5. Identification of 6-BA-Responsive Genes and GO Analysis

The RNA-sequencing reads were mapped to the *P. trichocarpa* genome (version 3.0) (http://www.phyozome.net/popalr.php) with TopHat (version 2.0.9) using the spliced mapping algorithm (multi hits ≤ 1). Cufflink (version 2.1.1) was applied to conduct gene quantification and difference analysis on mapping results of tophat. Expression levels of protein-coding genes were calculated and normalized using fragments per kilobase of transcript per million fragments (FPKM). The FPKM value was analyzed for differentially expressed genes among samples using the DEGseq software package. Fold change (multiple expression difference) and Fisher test were used to screen the difference degree of differentially expressed genes. The criteria of DEGs were as follows: (1) hypothesis test *p* value < 0.05; (2) fold-change ≥ 2; and (3) FPKM > 5 at all samples.

The online program AgriGo with Singular Enrichment Analysis (SEA) Tool (http://bioinfo.cau.edu.cn/agriGO/analysis.php) was used for GO analysis. The multitest adjustment method was used to perform the Fisher’ test, and *p* values were corrected by the false discovery rate (FDR). The minimum number threshold of mapping entries was set to 5.

### 4.6. Allele-Specific Expression Analysis

The RNA-sequencing reads were mapped to the *P. trichocarpa* genome (version 3.0) (http://www.phyozome.net/popalr.php) with TopHat (version 2.0.9) using the spliced mapping algorithm (multi hits ≤ 1). The mapping results were converted to compressed binary version of the Sequence Alignment Map(BAM) format, and non-unique and unmapped reads were filtered with SAMtools. The AddOrReplaceReadGroups and MarkDuplicates tools in the Picard package (http://broadinstitute.github.io/picard/, version: 1.87) were used for acquiring read group information, sorting, and marking duplicates. The HaplotypeCaller tool in Genome Analysis Toolkit (GATK, version: 3.8) was used for variation calling in the 6-BA and control groups. The minimum phred-scaled confidence threshold for calling variants was set at 20. CombineGVCFs tool in GATK was used to integrate variations in all samples. All other parameters were set at default values. To filter the resulting callset, the VariantFiltration tool in GATK was used to filter clusters of at least 3 SNPs within a window of 35 bases and low-quality SNPs. Annotation of variation between the samples from the 6-BA-treated and control groups was based on gene model set v3.0 from Phytozome v11.0 (Appendix A). Loci within 500 bp of the 3′ untranslated region (UTR) sequences of 6-BA-responsive genes with maximum support reads were selected as candidate loci to represent the ASE levels of genes. The ASE level was calculated based on the allele fraction (ALT/(ALT+REF)), where Alt refers to the number of reads that contain the alterative allele, and REF refers to the number of reads that contain the reference allele.

### 4.7. Predicting lncRNAs and Identifying 6-BA-Responsive lncRNAs

FASTX-Toolkit version 0.0.13 was used for quality control of RNA-seq reads, including the removal of low-quality reads and adapter sequences shorter than 20 nucleotides. The clean reads were aligned to *Populus* rRNA sequences using the Short Oligonucleotide Analysis Package 2 (SOAP2; http://soap.genomics.org.cn/soapaligner.html) to eliminate rRNAs. The clean data were aligned to the *P. trichocarpa* genome (version 3.0) with three base mismatches allowed. Cufflinks v2.1.1 [50] was used to assemble transcripts based on the *P. trichocarpa* genome (version 3.0). Expression levels were calculated and normalized using fragments per kilobase of transcript per million fragments (FPKM). The prediction of lncRNAs from RNA-seq data was performed following the method of Sun et al. [51].

### 4.8. Predicting Target Genes of 6-BA-Responsive lncRNAs

Potential target genes of the 6-BA-responsive lncRNAs were classified into two groups, *cis*-target genes and *trans*-target genes, according to their regulatory effects. Two different algorithms were used to predict the two types of target genes. The potential *cis* target genes, which are physically close to lncRNAs (within 10 kb), were predicted using genome annotation and a genome browser using the criteria described in Jia et al. [52]. Potential *trans*-targets were predicted by searching the *Populus* mRNA database based on mRNA sequence complementarity and RNA duplex energy prediction, assessing the impact of lncRNA binding on complete mRNA molecules. BLAST was used to select target sequences that were complementary to the lncRNA, setting *E*-value < 1e^−5^ and identity ≥ 95%. RNAplex software was used to calculate the complementary energy between two sequences for further screening and to select potential *trans*-acting target genes (RNAplex -e-60) [53].

### 4.9. Identification of 6-BA-Responsive 24-nt siRNAs

After filtering low-quality reads and adapters with FASTX (fastx_toolkit-0.0.13.2), clean siRNA sequence data were consolidated into an siRNA data set. The data set was further collapsed into FASTA format using a Perl script. Pln24NT (Pln24NT_analysis_local_version_v1.0) (http://bioinformatics.caf.ac.cn/Pln24NT/) was used to carry out siRNA analysis. The 24-nt siRNA read sequences were retrieved and aligned to miRNAs, and RFAM noncoding RNAs were removed with Perl scripts [54]. Non-redundant 24-nt siRNA sequences were mapped to the *P. trichocarpa* genome (version 3.0) for each dataset using Bowtie [55]: a maximum of one mismatch (-v 1) was allowed, and the best alignments of reads with no more than 50 hits (-a -m 50 -best -strata) were reported. SiRNA cluster calling was processed by ShortStack v3.3 with a minimum coverage of ten 24-nt siRNA reads. Genomic regions associated with siRNA clusters are referred to as siRNA loci hereafter. If present, 24-nt siRNA clusters were merged in a 150-bp window [56] to generate the final set of 24-nt siRNA loci. Transposon (TE) overlap analysis of clusters was conducted by comparing the start and end positions with TEs annotated by RepeatMasker (http://www.repeatmasker.org/) using the intersectBed and subtractBed tools in Bedtools [57]. Mapping 24-nt siRNA reads were selected for expression level analysis by calculating reads per million (RPM) for each locus. The normalized number of reads mapped to the genome of all 24-nt siRNAs from each 24-nt siRNA cluster (RPM, reads per million) represents the expression level of the cluster.

### 4.10. Genome-Wide Identification of 6-BA-Responsive Methylated Cytosine Loci and DMRs

Quality control reads of the reads was performed using FASTX-Toolkit version 0.0.13 (http://hannonlab.cshl.edu/fastx_toolkit/index.html) with default parameters. Specifically, high-quality sequencing reads (Q >30) from the 6-BA-treated and control groups were selected and trimmed to 101 nt and mapped to the *P. trichocarpa* genome (version 3.0) (http://www.phyozome.net/popalr.php). To detect methylation sites, the Bismark alignment tool was used to detect cytosines that had been converted after bisulfite treatment and to conduct methylation site calling with default parameters. Fisher’s exact test was used to test cytosine sites of the 6-BA-treated and control groups with a minimum coverage of ≥ 5.

The binomial distribution of the number of methylated cytosine and non-methylated cytosine frequencies at each site was tested to determine whether the site was a genuine methylated site. The methylation level (ML) of each identified methylated cytosine was calculated using the following formula: ML = mC/(mC + umC), where mC and umC represent the number of methylated cytosine and unmethylated sites, respectively. Due to the influence of bisulfite conversion rate, the methylation level was corrected using the following formula: ML_corrected = (ML-r)/(1-r), where r represents the bisulfite conversion rate. swDMR software was used to identify differentially methylated regions (DMRs). Based on the methylation information for each site (read coverage ≥ 5), this software uses a sliding window method to scan the genome and identify DMRs. The sliding window size was set to 1000 bp with 100 bp as a step size. Windows with probabilities < 0.05 were merged into larger regions to estimate the mean and variance of the entire methylation regions. ANOVA was used to filter DMRs with *p* < 0.05 (called candidate DMRs). To analyze the distribution of DMRs in various elements of genes, the DMRs were mapped to the *P. trichocarpa* genome (version 3.0) and annotated using Phytozome poplar genome v3.0 (http://www.phytozome.net/poplar.php). Chi-squared homogeneity tests were used to evaluate the effects on mapping distribution.

### 4.11. Genotyping of CpG Loci

A quantitative measure of the methylation level for each analyzed CpG site was assigned, with methylation support rate (S) ranging from 0 to 1.0, corresponding to no methylation on either allele to complete methylation of both alleles (S = number of mC reads/(number of mC reads + number of C reads in single locus).

### 4.12. Genome Re-Sequencing and SNP Calling

The SNP data used in this study were obtained by re-sequencing an association population composed of 435 unrelated *P. tomentosa* individuals. Total genomic DNA was extracted from individual leaves using a DNeasy Plant Mini Kit (Qiagen, Shanghai, China) according to the manufacturer’s protocol, followed by re-sequencing of the raw data at a depth of >15× using the Illumina GA II platform after constructing the library. Paired-end short reads (150 bp) were generated and filtered to obtain clean data by removing low-quality reads (≤50% of nucleotides with a quality score < Q20). The filtered reads were aligned and mapped to the *Populus* reference genome using BWA with default settings [58], with an 81% to 92% mapping rate and ~11× effective mapping depth for most individuals. Uniquely mapped paired-end reads underwent data quality control using fastqc software. BWA software was used for genome mapping of high-quality reads. After discarding the duplicate reads, SNP calling was performed using the HaplotypeCaller tool in GATK3.8 [3]. The genotype data for the DMRs were obtained using location information. SNP markers with minor allele frequencies <5% were excluded. SNPs with >25% missing data, as well as SNPs with limited numbers of individuals represented in the alternative genotype class or classes, were also discarded from the association analysis.

### 4.13. Single SNP-Based Association Analysis

Single SNP–trait association analysis was performed using the mixed linear model (MLM) in the software package TASSEL5.0 (https://www.maizegenetics.net/tassel) considering the effects of population structure (*Q*) and pairwise kinship coefficient (*K*). The pair-wise kinship coefficient (*K*) was evaluated with SPAGeDi 1.3 [59], and the *Q* matrix was calculated with STRUCTURE 2.3.4 based on significant subpopulations (*k* = 3) [60]. The parameters were used to correct the relationships of associated individuals to minimize false positives obtained by association analysis. The genotypic effects of associated SNPs were effectively decomposed into additive and dominant effects under this model. The positive false discovery rate (FDR) was calculated to correct for errors related to multiple testing using QVALUE software [61]; a *q*-value of 0.10 was selected as the significance threshold. SNP markers with a value of dominance to additive effects ratio (d/a) > 2 were defined as prominent dominant-effect loci.

### 4.14. Statistical Analysis

One-way ANOVA was performed using SPSS software (SPSS19, IBM Corporation, New York, USA), and significant differences between the 6-BA-treated and control groups were determined using Fisher’s least significant difference (LSD) test. Differences were considered statistically significant when *p* < 0.05. The correlation between the ratios of variation in DNA methylation and ASE was determined at a significance level of *p* < 0.01.

### 4.15. Data Availability

The RNA sequencing raw data and bisulfite sequencing raw data have been submitted to Genome Sequence Archive in BIG Data Center (BIG, CAS, China) under accession number of CRA002030. The raw data of genome re-sequencing had been deposited in the Genome Sequence Archive in BIG Data Center (BIG, CAS, China) under accession number of CRA000903.

## Figures and Tables

**Figure 1 ijms-21-02117-f001:**
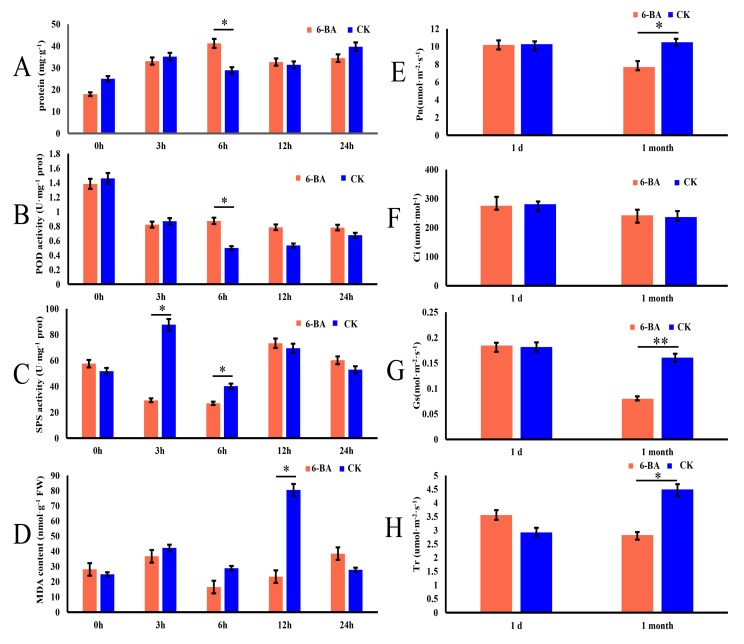
Effects of 6-benzylaminopurine treatment on physiological and photosynthetic characteristics of *P. tomentosa*. (**A**–**D**) Changes in total protein content, peroxidase (POD) activity, sucrose phosphate synthase (SPS) activity, and malondialdehyde (MDA) content in leaves at 0, 3, 6, 12, and 24 h of treatment. (**E**–**H**) Changes in net photosynthetic rate (Pn), intercellular CO_2_ conductance (Ci), stomatal conductance (Gs), and transpiration rate (Tr) measured 1 day and 1 month after treatment. Error bars represent standard deviation (SD) of three biological replicates (*n* = 3). Asterisks indicate significant differences between the 6-BA-treated and control groups (* *p* < 0.05, ** *p* < 0.01). The red bars represent the 6-BA treatment group (6-BA), and the blue bars represent the control group (CK).

**Figure 2 ijms-21-02117-f002:**
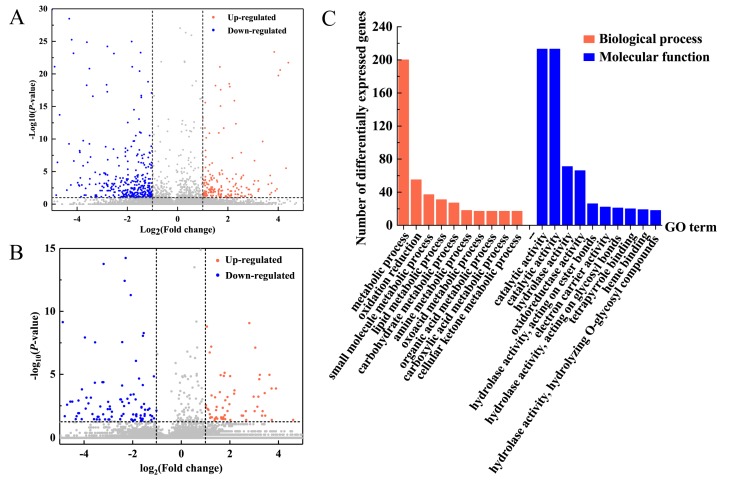
Analysis of 6-BA-responsive genes and lncRNA in *P. tomentosa*. (**A**) Differential expression levels (fold change) of 6-BA-responsive genes. (**B**) Differential expression levels (fold change) of 6-BA-responsive lncRNAs. (**C**) Functional categorization of 6-BA-responsive genes reveal enrichment in the biological process and molecular function gene ontology (GO) category.

**Figure 3 ijms-21-02117-f003:**
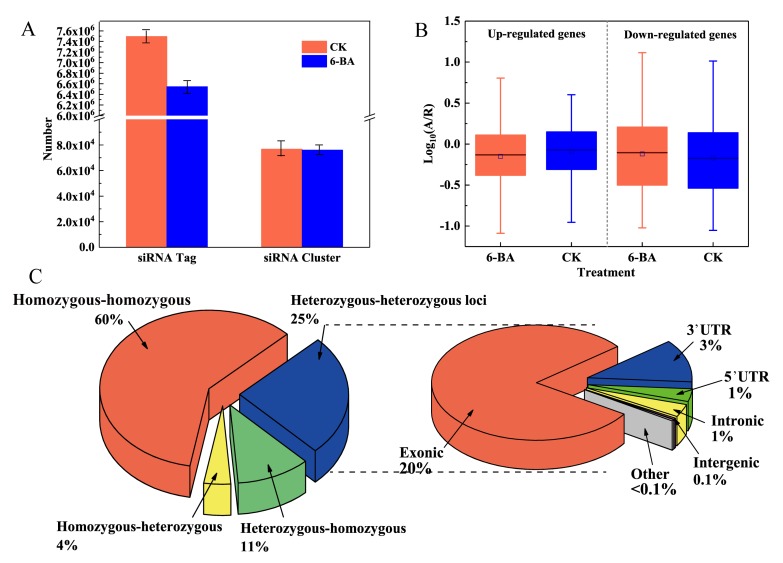
Analysis of 6-BA-responsive 24nt-siRNA and allele-specific expression in *P. tomentosa.* Changes in number and diversity of 24nt-siRNA under 6-BA treatment. (**A**) The numbers of reads that contain the aberrance allele, and R represents the numbers of reads that contain the reference allele. Error bars represent standard deviation (SD) of three biological replicates (*n* = 3). (**B**) Boxplot representing allele-specific expression levels of 6-BA-responsive genes in the CK and 6-BA treatment group, respectively. (**C**) Pie charts representing the classification of loci showing allelic variation under 6-BA treatment. Heterozygous–heterozygous loci represent the loci was heterozygous in CK and 6-BA groups; heterozygous–homozygous loci represent the loci was heterozygous in the CK group and homozygous in the 6-BA group; homozygous–heterozygous loci represent the loci was homozygous in the CK group and heterozygous in the 6-BA group; other represents the loci where no allele variation occurred, which was homozygous in CK and 6-BA groups (left). Distribution of allele-specific loci in different regions of the genome under 6-BA treatment (right).

**Figure 4 ijms-21-02117-f004:**
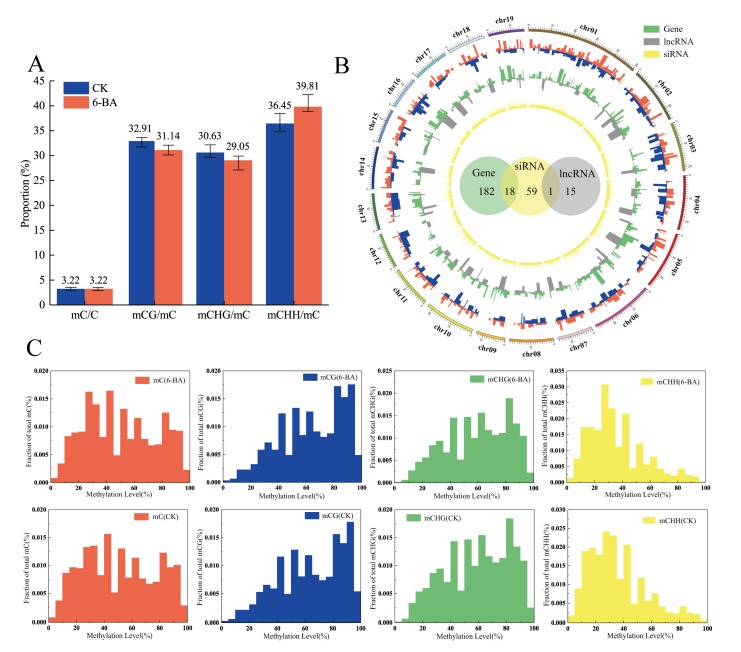
Patterns of variation in DNA methylation in response to 6-BA treatment in *P. tomentosa.* (**A**) Classification of methylated cytosines in the control (CK) and 6-BA-treated groups. Error bars represent standard deviation (SD) of three biological replicates (*n* = 3). (**B**) The outermost red ring represents the differential hypermethylation domains under 6-BA treatment; the blue ring represents the differential hypomethylation domains under 6-BA treatment; the green ring represents differentially expressed protein-coding genes within DMRs boundaries; the gray ring represents differentially expressed lncRNAs within DMR boundaries; the innermost yellow ring represents differentially expressed 24-nt siRNAs within DMR boundaries. The green, yellow, and grey circles in the center represent reliable, differentially methylated transcriptional elements, and the overlapping region represents the overlap between two kinds of transcriptional elements. (**C**) Distribution of methylation levels in different contexts in the control (CK) and 6-BA-treated groups.

**Figure 5 ijms-21-02117-f005:**
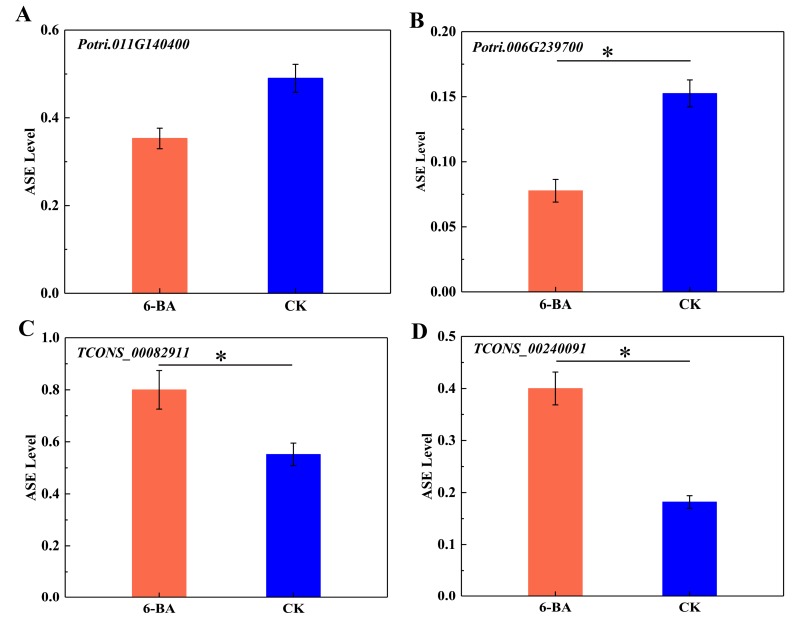
Analysis of allele-specific expression in a single gene in response to 6-BA treatment. (**A**) Variation of ASE levels in *Potri. 001G140400*. (**B**) Variation of ASE levels in *Potri.006G239700.* (**C**) Variation of ASE levels in lncRNA (*TCONS_0082911*). (**D**) Variation of ASE levels in lncRNA *TCONS_00240091*. Error bars represent standard deviation (SD) of three biological replicates (*n* = 3). Asterisks indicate significant differences between 6-BA-treated (red) and control (blue) groups (* *p* < 0.05).

**Figure 6 ijms-21-02117-f006:**
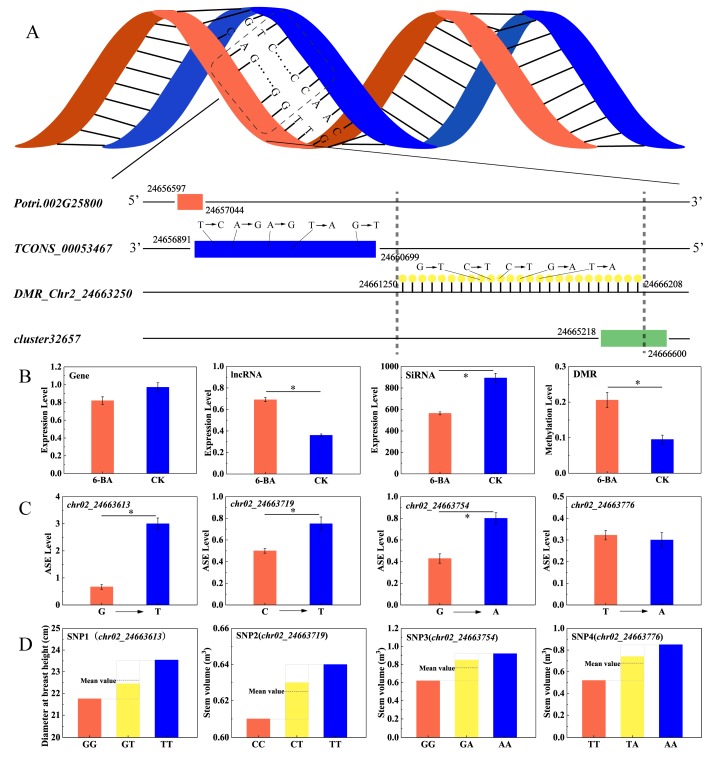
Analysis of patterns of genetic variation in candidate DMR regions and interactions between genomic elements. (**A**) The gene and lncRNA are located on complementary DNA strands, and a DMR partially overlapping with a siRNA cluster occurs in the upstream region of lncRNA. (**B**) Variation of expression levels of *Potri.002G258000*, *TCONS_00053467, siRNA_cluster_32657,* and variation of DNA methylation level of *DMR_Chr02_24663250*. (**C**) Variation of ASE levels in SNPs located in lncRNA (TCONS_0053467). (**D**) Differences between the phenotypic values of the heterozygous genotype and the mean value of the homozygous allele indicate dominant effects. Error bars represent standard deviation (SD) of three biological replicates (*n* = 3). Asterisks indicate significant differences between 6-BA-treated and control groups (* *p* < 0.05).

**Table 1 ijms-21-02117-t001:** SNPs of DMRs associated with *P. tomentosa* growth and wood property traits.

Trait	Number of SNPs
Stem volume (V, m^3^)	46
Diameter at breast (D, cm)	107
a-cellulose content (Ac, %)	2
Hemicellulose content (HEMC, %)	4
Holocellulose content (HC, %)	11
Lignin content (LC, %)	1
Fiber width (FW, μm)	11
Total	182

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
