# Peer review of "Changes in DNA Methylation in Response to 6-Benzylaminopurine Affect Allele-Specific Gene Expression in Populus Tomentosa"

_ijms, 2020, doi:10.3390/ijms21062117_

Round 1

Reviewer 1 Report

The author presented an impressive amount of data from various source (phenotyping, RNA-seq, BS-Seq, genome resquencing, biochemical measures, etc.) that they tried to correlate together in order to investigate the effects of 6-BA on poplar. The analytical work is impressive, but I find, in general, the presentation of the results to be difficult to access for the common scientific reader. 

First of all, I believe that the manuscript lacks of a central thread to follow. I find difficult to link all the analyses together in order to answer a main question, and I tend to get lost in the huge amount of data that is presented. Overall, it is difficult in both the text and the figure to be able to quickly spot the most relevant information. On that aspect, I would suggest the authors to maybe limit the number of figures and data presented in the result section. I am not convinced that they are all very informative or essential for the reader. As a lot of correlation are presented (between DRM, ASE, up/down regulation), maybe Venn diagrams could also help to quickly visualize the data?

Still reading the result section, I very often found myself unable to clearly understand how the data was generated or analyzed, and had to go back and forth from the methods to the results sections. Some analyzes (ex: the ASE analysis) took me until the very end of the paper to understand how they were performed. A bit more description of how the data was obtained during the presentation of the results would really help.

As the article is mostly descriptive and correlative, I believe it is crucial to rearrange the way the data is presented (and maybe filter it a bit) to make it easier for the reader to get the big picture. 

General comments

All abbreviations should be explained in the text the first time they are mentioned. Too often, abbreviations are explained only in figure legends or sometimes not explained at all. This is very frustrating for the reader (example lines 99-101, figure 3d, line 454, line 615 etc.)

I feel that a part where you would present the result of differentially expressed genes between controls and 6-BA plants is missing. You basically do it for lncRNA, but not for protein coding genes. I would be interested to see how much the transcriptome is affected by the treatment. 

I struggled a lot to clearly understand how the ASE analysis was done, and I still have a lot of questions about it. Was it done only on differentially expressed genes between controls and 6-BA, or on absolutely all genes? In the latter case, do you see a lot of ASE without difference in gene expression? Also, on figure 4, it looks like you only focus on genes where you can find heterozygous SNP in the RNAseq data of both controls and 6-BA plants. Why did you remove homo-hetero and hetero-homo genes? They compose 15% of your data and seem to carry valuable information to me. If your plants are really clonal and these genes truly heterozygous, it would mean that in some conditions, only a single allele is transcribed. I think this should be discussed further.

Specific comments

Lines 177-180: How do you explain the huge drop in amount and diversity of siRNA in 6-BA treated plants? Any hypotheses?

Lines 99-102: as a biologist not very familiar in plant physiology, I would have liked a very brief explanation of why these physiological parameters where chosen as a measurement of the 6-BA effect.

Figure 1: please explain “CK” in the legend. 

Lines 147-148: use either mean or median for both (I would recommend using median).

Figure 2b: the size of the text under the histogram is too small. 

Line 156: what are the enriched GO terms in biological process? Could it be presented in supplementary material?

Figure 3 a-c: Reading the figure would be easier if you could write the data source (Prot. Cod., lncRNA, 24 nt siRNA) next to the heatmaps.

Figure 3d: Please explain in the legend what A/R stands for.

Line 194-195: What is the average heterozygocy for P. tomentosa? Here you say that only a small proportion of the genome is heterozygous, although you find that 19,200 genes contain SNPs, out of around 41 000 genes in P. trichocarpa. It means that around 1 gene out of 2 has at least 1 SNP in its transcribed sequence. I find it to be an important information.

Lines 199-200: how do you define your loci here? Do you mean heterozygous SNPs?

Lines 201: 78.19% are located in CODING exons. UTRs are also exons. 

Lines 204-205: In this part, it was not clear to me if the A/R ratio was calculated for each gene (doing a sum of all the SNPs in the gene) or individually for all SNPs positions. I found the answer in the methods but a short explanation here would help the reader

Figure 4a: I suggest you use “homozygous-homozygous” instead of “other”

Line 227: You state that the BS-mapping was done on P. trichocarpa genome. As a plant scientist not working on Populus, I would have like to know here to which extent these genomes are similar (or different), and what bias it could cause. 

Lines 228-231: I am quite surprised by the low % of uniquely mapped reads. From my knowledge, ~40% of the genome of P. trichocarpa are repetitive elements. How do you explain this low number of uniquely mapped reads? Is it because of the differences between P. tomentosa and P. trichocharpa genomes?

Lines 239-243: You suggest that the increase in CHH methylation in 6-BA treated plants may be related to 6-BA responsive siRNA. Isn’t that counter-intuitive, as 99.76% of the siRNA were downregulated in 6-BA plants?

Line 251: Which threshold was used to make the 2 groups based on the S variation rate?

Lines 251-255: What is the overlap between the ASE loci and the differentially methylated loci? It is not clear to me how this was done. Did you consider the CpG loci when they were inside the ASE gene coding sequence or whole sequence, or when they were near the gene, etc? 

Lines 261-265: How are these DMR distributed? Are they mostly in exons, introns, promoters, intergenic, etc? Is there a pattern?

Figure 5A: I am surprised that you have no error bar on this figure. Was there absolutely no variation between the replicates?

Figure 5b: this figure would gain to be bigger 

Line 293: fold change values or log(fold change)?

Lines 304-306: In this part, you look at the link between ASE and up/downregulation but you never mention if these genes, that you state are in DMRs, are hyper or hypomethylated in 6-BA when compared to the controls. Also, can you generalize your interpretation regarding the fact that the ASE in protein coding genes decreases in 6-BA (among these 210 protein-coding genes located in DMRs) and the opposite in lncRNA? Or this just apply to those 2 genes and 2 lncRNA?

Lines 320-322: what criteria did you use to choose these 507 DMR out of the 566 that you detected?

Line 430: what is your threshold to define “high numbers of reads”?

Lines 437-439: I may be completely wrong here, but I thought that most regulatory elements would be in 5’ of the genes, not in 3’. Could you please add a reference to this claim?

Lines 444-445: I find the affirmation here to be speculative

Line 592: Did you also use SOAP2 here?

Author Response

Dear Dr. Reviewer:

Reviewer 2 Report

The manuscript from Anran Xuan et al present omics data in populus tomentosa of cytokinin treated leaves 6 hours post treatment. There is actually an important challenge to better understand the relationship between epigenetics notably DNA methylation and hormonal control. The present paper is totally relevant in this frame as proposing transcriptomic data (coding and non coding) as well as DNA methylation. The manuscript is well presented and is associated to many data and supplementary files.

   However, I have major concerns about the experimental design and the methodology of omic analysis that need to totally revised the paper before establishing an adequate set of data that could be interpreted and publish.

  1. It is not clear which leaves (age, rank???) are sprayed with CK (or water… do CK is dissolved in water?)? and how spraying water to leaf surface can allow hormone to enter? References to this methodology? Concentration used here compare to internal amount? Experimental evidence that CK is entering leaf tissue? And how much leaves are sprayed and how much are pooled??? The experimental design is lacking.
  2. Looking to bibliography, I find bibliography about a methodology doi:10.3390/biom9010012 and  https://doi.org/10.3389/fpls.2017.00493. Al least here authors must give similar details of procedure to justify their methodology.
  3. Can authors justify the choice of leaf? How can they connect their effects measure 24 hours or 1 month after spraying and propose to analyse omics at 6hours?
  4. Can author give support to the fact that 6hours post CK, it is possible to measure on leaf event of DNA methylation on hundreds of loci….. Without any (or about cell division) ? Any experimental support to the immediate (6 hours is really very short) effect on DNA methylation at the genome scale in non-dividing leaf tissues? I am not sure there is any other experimental support to such as short time scale.
  5. The previous remark is associated to strong doubts about the bioinformatics and statistical thresholds that have been applied here and that are globally of very low quality. For transcriptomics to avoid to analyze with only 3 repeats genes with very low differential of expression, it is not recommended to use a FPKM >5 in a t least one sample to select DEGs that could have significant biological variations. Here when looking to Figure 3, it seems that all DEGs are of very weak differential and expression level…. That can overestimate the biological significance. For methylome analysis the criteria used are too low quality. First only 25 M reads are mapped. It means that there is a theoretical cover of 10x… this is very low and at least 20x is recommended. For reads quality Q>30 is recommended (and not 20), minimum coverage for cytosine is usually over 5… 10 is better. DMR must be associated to a minimum of methylation differences of 10% (for CHH) to 25% (for CG and CHG) to avoid to select DMR with too low differential of methylation and no biological relevance. DMR must be at least >100pb (and not 23 as said in result section). Analysing DMRs with very different size is also not recommended (23 to 1895 is a too large range of size). I am afraid that with this new criteria… the number of DMR will be low or nothing… and that is maybe related to the only 6hours post treatment design.  
  6. The thematic is about poplar and DNA methylation… but the authors do not cite the actual bibliography on that topic! This is exactly the same for the bibliography about connection between epigenetics and hormones. This is necessary for the discussion.
  7. Last part of the introduction is not clear… L80-94… this is a mix of sentences without clear connections. Reformulate.
  8. Some conclusions have no support: L139-141.
  9. Figure 1 is a mix of several traits without connections and time scale compare to the omic analysis done at 6hours.
  10. Figure 2 GO analysis must be treated by GO enrichment procedure with a statistical analysis and not list of GO terms. The Figure 2B is not readable.
  11. Fig4B I do not understand the red line ? you try linear correlation with this points? Did you check significance and possibility to do that?
  12. All result and discussion must be rewritten at the view of new transcriptomic and methylome data.

Author Response

Dear Reviewer,

Round 2

Reviewer 1 Report

Dear authors,

Thank you for submitting this reviewed version of the article. I also thank the authors for their very detailed and documented answers to my questions. I found that the new presentation and modifications make the article easier to read and to aprehend. 

minor corrections
Line 134: we sprayed plants with A 6-BA..
Line 312: Our experimental ... (remove Due to)
Line 319: AN obvious

Author Response

Response to Reviewer 1

  1. “Line 134: we sprayed plants with A 6-BA.”

Our response: We thank Reviewer 1 for this suggestion. “A” has been deleted.

  1. “Line 312: Our experimental ... (remove Due to)”

Our response: We thank Reviewer 1 for this constructive suggestion. “Due to” has been removed.

  1. “Line 319: AN obvious”

Our response: We thank Reviewer 1 for this suggestion. It has been corrected

Reviewer 2 Report

The manuscript has been really improved and several concerns have been well treated.

Even, if I have still doubt about the possible effect at  hours post treatment on DNA methylation, the authors have revised their methodologies and I think it is a good procedure now. However they do not provide any rational and other published work showing that their observation are not artefactual at 6hours for DNA methylation. This must be at least clearly discussed in the discussion part. ... and propose future analyses that could support this work. 

However, I was strongly disapointed by the text revision for several points. While the authors say in the response to reviewezrs letter that they have done modifications on the manuscript, it is not the case or it is totally unsufficient.

For example, I ask that the avaiblable bibliography on poplar methylome as well as connection with hormones should be cited....only one paper was discussed : Raj et al  2011... and this is totally unsufficient and was wrong. Thus Raj et al 2011 publish a paper on global DNA methylation on leaves... and here they speak about DNA methylation in SAM.... so they confuse with Gourcilleau et al 2010 that is cited in Raj et al 2011. 

The discussion is said by the authors to be revised ... but only few sentences have been modified... the same for introduction.

I think that the paper is really improved by the significance of the work ... and the text particularly for discussion must be edited again before publication. 
